# Evaluating Surface Water Nitrogen Pollution via Visual Clustering in Megacity Chengdu

**Yao Ding** [1,2,*] , **Yin Wang** [1,2], **Shuming Yang** [1,2], **Xiaolong Zhao** [1,2], **Lili Ouyang** [3] **and Chengyue Lai** [3]

1    Southwest Municipal Engineering Design & Research Institute of China, Chengdu 610084, China;
    wangyin@ccccltd.cn (Y.W.); yangshuming@ccccltd.cn (S.Y.); zhaoxiaolong1@ccccltd.cn (X.Z.)
2    Research and Development Center on Urban-Rural Water Environment Technology (CCCG),
    Wuhan 430058, China
3    Institute of Water Environment, Chengdu Institute of Environmental Protection, Chengdu 610072, China;
    ouyangll@cdaes.org.cn (L.O.); laicy@cdaes.org.cn (C.L.)
*    Correspondence: dingyao1220@foxmail.com; Tel.: +86-18000578970

**Abstract:** The current standards used for nitrogen pollution evaluation are lacking, and scientific classification methods are needed for nitrogen pollution to improve water quality management capabilities. This study addresses the important issue of assessing surface water nitrogen pollution by utilizing two advanced multivariate statistical techniques: self-organizing maps (SOMs) obtained using the K-means algorithm and the Hasse diagram technique (HDT). The research targets of this study are the rivers of the megacity Chengdu, China. Samples were collected on a monthly basis in 2017–2020 from different sites along the rivers, and their nitrogen pollution parameters were determined. The grouping of nitrogen pollution parameters and the clustering of sampling events using SOMs facilitate the preprocessing required for the HDT, wherein clusters are ordered according to the pre-clustered water sampling events. The results indicate that nitrogen pollution in the Chengdu River Basin, which is prominent and mainly driven by nitrate nitrogen, can be categorized into five levels. The nitrogen pollution in Tuo River is serious. Although the degree of ammonia nitrogen pollution in Jin River is higher, the pollution range is smaller. Furthermore, these results were evaluated by the SOMs and HDT to be clear and reliable. Overall, these findings can provide a basis for local environmental legislation.

**Keywords:** megacity; self-organizing maps; Hasse diagram technique; nitrogen pollution; environmental management



## 1. Introduction

Theoretical and experimental advances in water environment quality, which is a well-known indicator of the degree of pollution, are vital for protecting the water ecological environment [1,2]. Earlier evaluations of water environment pollution have mainly been performed using qualitative descriptions of water. An extensive understanding of the physical, chemical, and biological effects of the water environment has been obtained over the years using several water quality evaluation methods such as index evaluation [3,4], fuzzy mathematics theory [5], grey system theory [6], multivariate statistical analyses [7–10], and artificial neural networks [11–13] . Owing to the rising pressure on water quality management objectives, there is an urgent need to analyze data and obtain important information; however, this has become difficult due to an increase in the historical monitoring data and automatic station data. Accordingly, the need for scientific and efficient water pollution assessment methods has arisen. Therefore, the research and application of artificial neural networks and Hasse diagram technology (HDT) have become a future development trend.

Self-organizing maps (SOMs) were first pfroposed by Finnish scholar Kohonen in 1982 [14]. As a nonlinear science, SOMs have the advantages of autonomy and inclusiveness. However, since clustering results cannot be used to compare each SOM individually,

their practical applicability for environmental management is limited. HDT, which has been named after the German mathematician Helmut Hasse, is a method based on the partial order set theory that retains the important elements in the evaluation and decision-making processes [15,16]. This method only requires the weight order of the evaluation index, thus circumventing the need to weigh in other water quality evaluation methods. However, HDT exhibits high intolerance to 'noise'; thus, it has high requirements for data preprocessing. Although SOM and HDT have been used together for river pollution assessments, insufficient information has been obtained. Li et al. [17] only used two methods to evaluate water pollution independently, while limited information was interpreted using complex Hasse images. Meanwhile, Voyslavov et al. [18,19] and Liu et al. [20] only used SOMs for parameter grouping, and the equivalence class division of samples still relied on local surface water quality standards.

According to most global standards, rivers require only limited total nitrogen (TN) concentrations; however, these standards lack the concentration requirements for various other nitrogen forms. According to the surface water quality standard in China (GB3838-2002), river water is evaluated only using $NH_3$-N. Meanwhile, lakes and reservoirs are evaluated using TN and $NH_3$-N. Although the mass concentration of $NO_3^-$-N is limited in drinking water ($\leqslant$10 mg/L in China), it exhibits a wide range. Traditional analytical methods offer a more qualitative description, which is insufficient for evaluating nitrogen pollution in rivers.

Under the absence of standards, this study used SOM and HDT techniques to explore the characteristics of regional nitrogen pollution and classify the river water pollution in Chengdu. In this study, no river water quality standard has been used as a reference except for the $NH_3$-N concentration. Therefore, SOM is used to simultaneously categorize the equivalence classes of parameters and samples, thereby eliminating the need for manual classification and successfully completing the 'noise reduction' processing of data. Finally, a concise and clear Hasse diagram is obtained, and the nitrogen pollution of samples is ranked. Based on the binomial results, the spatial and temporal distribution laws of large data set elements are determined. Overall, the advantages of both SOMs and HDT have been exploited, while their shortcomings have been addressed.

The study aims to offer chemometric expertise for comprehensively evaluating the nitrogen pollution in the river waters of Chengdu and provide a basis for local environmental legislation.

## 2. Materials and Methods

### 2.1. Study Area

The Yangtze River is China's 'mother river', and the Yangtze River Economic Belt is a major engine for China's development [21]. Chengdu is the nearest megacity to the Yangtze River Basin, and its water quality directly restricts the economic development and water safety in the lower reaches of the Yangtze River. It is located between 30°05′ N and 102°54′ E, has a population of 20.9 million, and covers an area of 14,335 km$^2$. Furthermore, it is positioned within the subtropical humid monsoon climate zone, and experiences an annual rainfall of 800–1400 mm and an average annual temperature of 15.2–16.6 °C [22]. Land use types in Chengdu City have the following three characteristics: First, land types are diverse. Second, the plain area accounts for 40.1% of the city area. Third, the land reclamation index (38.2%) is higher than the national average (10.1%). The area of construction land in Jin River Basin is 483.56 km$^2$, which is higher than that in Jinma River Basin and Tuo River Basin. The area of agricultural land in the Jinma River Basin and the Tuo River Basin is 3333.25 km$^2$ and 4749.67 km$^2$, respectively, which is significantly higher than that in the Jin River Basin.

Chengdu straddles two water systems: the Min River and Tuo River. The Min River, which was once considered the Yangtze River's main tributary, is divided into the Jinma River Basin and Jin River Basin at the Dujiangyan Fish Mouth (i.e., part of a famous ancient water project). Since ancient times, fish mouths have provided a steady flow of water

to Jin River throughout the year, thus facilitating agricultural irrigation and preventing floods. Excess water tends to flow toward the Jinma River, which is mainly used for flood discharge. Although the Tuo River has its own water system, it actually draws water from the Min River. Notably, the Jinma, Jin, and Tuo River Basin account for 44.43%, 15.94%, and 39.63% of the total watershed area, respectively [22].

## 2.2. Sample Collection

This study used 75 sampling points (Figure 1) in Chengdu River Basin, and 891 annual average values were collected between 2017–2020.

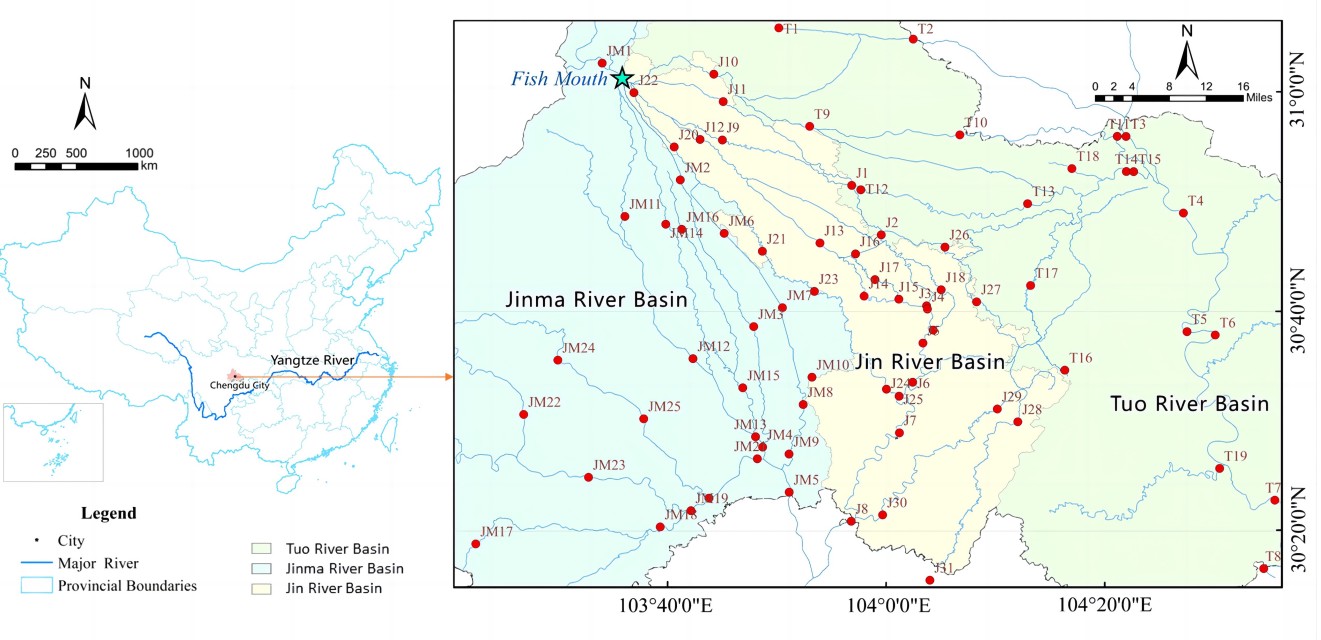

**Figure 1.** Location of sampling points.

The first working day of each month was used for sampling, and 21 physical and chemical indicators (flow, pH, DO, temperature, EC, $COD_{Mn}$, $BOD_5$, TN, $NH_3$-N, $NO_3^-$-N, $NO_2^-$-N, DON, TP, $PO_4^{3-}$, $K^+$, $Na^+$, $Ca^{2+}$, $Mg^{2+}$, $Cl^-$, $SO_4^{2-}$, $CO_3^{3-}$) of the samples were tested. Furthermore, the annual average concentrations of total nitrogen (TN), ammonia nitrogen ($NH_3$-N), nitrate nitrogen ($NO_3^-$-N), nitrite nitrogen ($NO_2^-$-N), and total organic nitrogen (DON) at the 75 sampling sites from 2017 to 2020 were used. These indexes were analyzed after the water samples were filtered in situ with disposable filter devices (0.45 μm pore size, 25 mm diameter, Whatman, GD/X, Maidstone, UK), frozen, and stored at <4 °C in centrifuge tubes made of polyethylene terephthalate (15 mL, sterile, Corning, NY, USA). $NO_3^-$-N and $NO_2^-$-N concentrations were measured using ion chromatography (883 Basic IC, Metrohm, HeriSau, Switzerland), while the $NH_3$-N concentrations were determined using spectrophotometry (722N, Shanghai Jingke, Shanghai, China). The TN concentration was digested using alkaline K persulfate and analyzed via spectrophotometry (UV752, Shanghai Jingke, Shanghai, China) after reducing $NO_3^-$-N to $NO_2^-$-N. Meanwhile, DON is calculated as follows: DON = TN-DIN = TN-$NH_3$-N-$NO_3^-$-N-$NO_2^-$-N [22,23].

## 2.3. Chemometrics

### 2.3.1. SOMs

SOMs are a neural network model used for exploring and visualizing high-dimensional data sets in the environment. Based on the minimum criterion of the Davies-Bouldwin index (DBI), this study uses K-means clustering for the automatic generation of final clustering categories [19,20]. Thus, this method can provide variable distribution information of the data sample by outputting variable planes. Furthermore, the K-means algorithm of

SOM can also output the unified distance matrix (U-matrix), which governs the construction of SOMs according to the distance between nodes and obtains the classification results of all nodes. The difference between the U-matrix and variable plane is that it includes all the variable information of the samples. The SOM clustering analysis was conducted using the SOM toolbox 2.0 in MATLAB 2021b software.

2.3.2. HDT

HDT is a data graph that can represent finite posets. According to the research results of Voyslavov et al. [18,19] and the user manual associated with Decision Analysis by Ranking Techniques (DART) [24], the steps required for HDT clustering are briefly explained:

(1) First, the weight order of each index parameter is determined. The calculation method of entropy weight is as follows [25]:

$$X = \left[ X_{ij} \right]_{n \times m} = \begin{bmatrix} x_{11} & x_{12} & x_{13} & \cdots & x_{1m} \\ x_{21} & x_{22} & x_{23} & \cdots & x_{2m} \\ x_{31} & x_{32} & x_{33} & \cdots & x_{3m} \\ \vdots & \vdots & \vdots & \ddots & \vdots \\ x_{n1} & x_{n2} & x_{n3} & \cdots & x_{nm} \end{bmatrix} \tag{1}$$

For $n$ samples and $m$ indicators, $X_{ij}$ is the value of the $i$th sample corresponding to the $j$th index.

(2) Calculate the normalization matrix:

$$X_{new} = \left| \frac{X - X_{min}}{X_{max} - X_{min}} \right| \tag{2}$$

$$N = \left[ X_{ij} \right]_{n \times m} \tag{3}$$

(3) Calculate entropy for all criteria:

$$\rho_{ij} = \frac{X_{ij}}{\sum_{i=1}^{n} X_{ij}}, \cdot (i = 1, 2, \ldots, n; j = 1, 2, \ldots, m)$$
$$e_j = -k \sum_{i=1}^{n} \rho_{ij} \cdot \ln(\rho_{ij}), (i = 1, 2, \ldots, n; j = 1, 2, \ldots, m) \tag{4}$$

where $\rho_{ij}$ is the weight of the $j$th sample value in the $i$th index, $e_j$ is the entropy of the $j$th index, and $k$ is the Boltzmann constant ($k = 1/\ln(n)$, ($0 \leqslant e_j < 1$)).

(4) Calculate the entropy weight $w_j$ of the $j$ indicator:

$$W_j = \frac{1 - e_j}{\sum_{j=1}^{m} d_j}, (j = 1, 2, \ldots, m) \tag{5}$$

Thus, the value of $W = (w_1, w_2, w_3, \ldots, w_j$ can be obtained ($\sum_j^n w_j = 1$).

Second, the Hasse matrix is obtained using HDT. The ranking of object $E$, which includes the sampling data of the research period, is performed based on variables such as the selected water quality parameters; this object is called Information Basis (IB). The processed data matrix $Q(N \times R)$ contains $N$ objects and $R$ variables. $y(x)$ represents the numerical value of the $r$th variable, and $y_r$ indicates the variables according to which the objects are ranked. The two objects $s$ and $t$ are comparable in the following cases:

$$s, t \in E; s \leq t \leftrightarrow y(s) \leq y(t)$$
$$y(s) \leq y(t) \leftrightarrow y_r(s) \leq y_r(t), \forall y_r \in IB \tag{6}$$

Even if one $y(s) \leqslant y(t)$, the objects $s$ and $t$ cannot be compared. The Hasse matrix, which can easily derive the partial order set and determine the relations between objects, can be expressed as follows:

$$h_{st} \begin{cases} +1 & \text{if} \quad y_r(s) \geq y_r(t), \forall y_r \in IB \\ -1 & \text{if} \quad y_r(s) < y_r(t), \forall y_r \in IB \\ \quad 0 & \text{otherwise} \end{cases} \tag{7}$$

Finally, the Hasse image is drawn according to the Hasse matrix. If there is no object $a$ in $E$, for which $s \leq a \leq t$ $(a \neq s \wedge a \neq t)$, $s$ is covered by $t$ or vice versa. The order relation in the Hasse matrix can be represented using the Hasse diagram, which is constructed as follows:

a. Each object or equivalence class has a circular representation with an identifier. The equivalence elements function as different objects, indicating that all variables in IB have the same value.

b. If there is a coverage relationship, the corresponding objects are connected by lines and the representative elements can be compared.

c. If $s \leq t$, $s$ is drawn above or below $t$; all the relation lines follow the same direction principle.

d. If $s \leq t \wedge t \leq z$, $s \leq z$. Although there is no connecting line between $s$ and $z$, a straight line can be used to connect $s$ and $t$.

e. If $s \leq t \vee t \leq z$, $s$ and $t$ are not comparable and cannot be connected using a straight line.

Elements that are not covered by other objects are termed as 'maximal elements', and those not covered by other objects are 'minimal elements'. Meanwhile, 'chain' and 'anti-chain' represent a set of comparable and incomparable objects at the same level, respectively; that is, the graph height represents the longest chain, and the graph width represents the longest anti-chain.

Since HDT is not tolerant to 'noise', preprocessing steps are extremely important. In this study, SOMs were used to preprocess the data, and HDT is implemented using the DART software [26].

## 3. Results

### 3.1. SOM Clustering Results

3.1.1. Determining the SOM Clustering Structure

In this study, the multi-year average of 75 monitoring sections for 12 months (a total of 891 samples) was used as the data set. According to the minimum node volume of the competition layer ($5 \times \text{INT}(\sqrt{N})$), the number of neurons in the SOM map was determined as 150 and statistical calculations were performed according to the data analysis method in Section 2.3.1. Figure 2a shows the U-matrix of the input dataset and visualizes all the parameters. The distance between neurons can be reflected by the U-matrix to determine the clustering structure of the SOM graph. The attribute value of the index parameters corresponding to each neuron can be expressed using color depth. That is, the neurons with higher TN and $NH_3$-N values were located in the upper and middle parts of the SOMs, and the neurons with higher $NO_3^-$-N, $NO_2^-$-N and DON values were located in the lower right part of the SOMs. Figure 2 shows that some neurons were not only polluted by $NH_3$-N, but also by $NO_3^-$-N, $NO_2^-$-N, and DON.

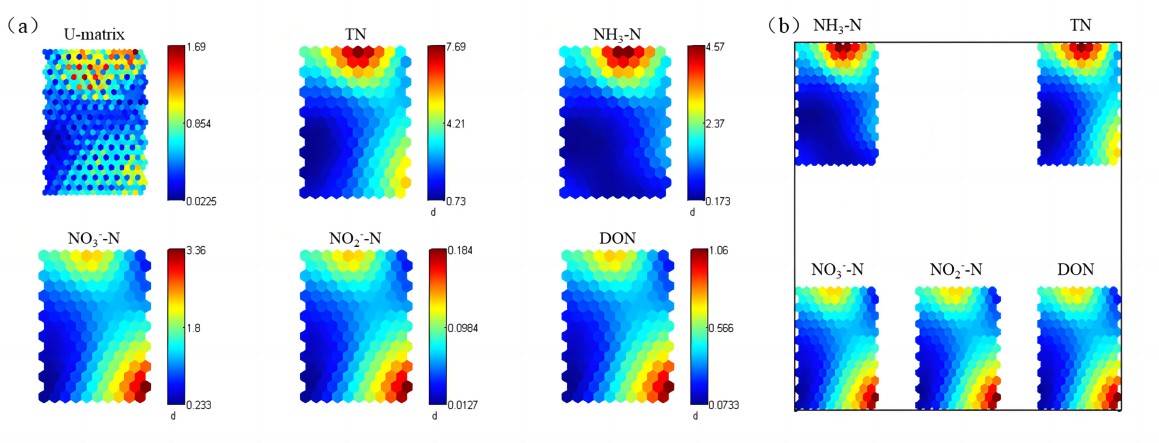

**Figure 2.** (**a**) U-matrix and variable planes for the input data, and (**b**) ordering of component planes.

### 3.1.2. Evaluation Index Selection

The plane ordering of water quality parameters is shown in Figure 2b, which also depicts the position, distance, and color of each parameter on the graph. Three distinct groups can be observed; the first group includes $NH_3$-N, the second group comprises TN, and the third group contains $NO_3^-$-N, $NO_2^-$-N, and DON. The images of the parameters in the third group show a high degree of consistency, indicating that there is a significant correlation between them. $NO_3^-$-N is the main form of nitrogen in river water and is more representative than $NO_2^-$-N and DON; thus, $NO_3^-$-N represents $NO_2^-$-N and DON to be a group. TN, $NH_3$-N, and $NO_3^-$-N parameters exhibit distinct distributions, thereby providing different information for data set objects. Therefore, TN, $NH_3$-N, and $NO_3^-$-N were selected as the evaluation indexes for water nitrogen pollution assessment based on HDT.

### 3.1.3. SOM Clustering Results

In this study, 891 objects were distributed in 142 neurons, and 8 neurons were not filled with objects (Figure 3d). Finally, the data samples were divided into 8 clustering categories (Figure 3a) denoted as $C_i$ ($i$ = 1, 2, ..., 8). Different cluster categories in Figure 3b correspond to distinct color partitions, with the corresponding number representing the cluster category ($i$). Figure 3c indicates the corresponding neurons in different clustering categories. Neurons numbered 1 to 150 are filled in order from left to right and from top to bottom. Figure 3d shows the number of samples contained in each neuron. For example, $C_1$ contains 11 neurons (119, 120, 13, 133, 134, 135, 146, 147, 148, 149, and 150) and a total of 78 samples.

### *3.2. HDT Clustering Results*

#### 3.2.1. Determining the Data Set Equivalence Class and Evaluation Index Weight Ranking

To reduce the irrelevant differences between objects, each filled node in SOM has been used as an equivalence class. Therefore, 891 objects are included in 142 neurons, and these neurons are then divided into 8 categories according to the water quality characteristics between nodes. These categories are used as the final equivalence class for HDT clustering analysis. When dividing the equivalence class of the data set, it is necessary to consider the weight ranking of the evaluation indicators. According to the selection results of the evaluation indicators in Section 3.1.2 and the methods described in Section 2.3.2, the weights of the evaluation indicators are calculated (Table 1).

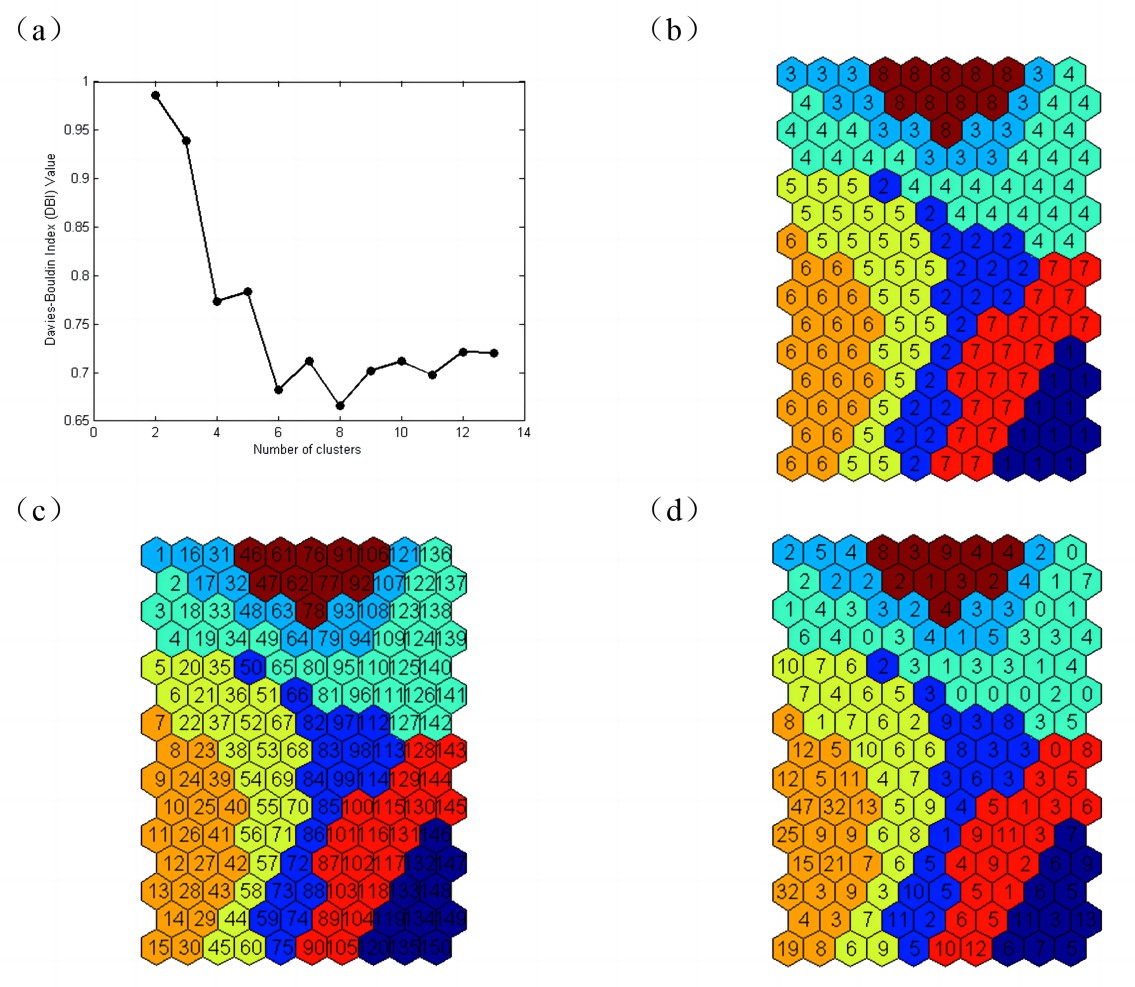

**Figure 3.** SOM clustering results: (**a**) relationship between the clustering number and DBI index, (**b**) clusters based on the lowest DBI index, (**c**) neuron numbers, and (**d**) number of samples in each neuron.

**Table 1.** Entropy weight of evaluation indices.

| Name | TN | NH$_3$-N | NO$_3^-$-N |
|---|---|---|---|
| $W_{ij}$ | 0.2087 | 0.4079 | 0.3834 |
| ranking | 3 | 1 | 2 |

### 3.2.2. HDT Clustering Ranking

The preprocessing results of data sets and evaluation indicators are input into the DART software, after which the Hasse diagram is output (Figure 4). The input object is divided into five levels (clean, generally clean, lightly polluted, moderately polluted, and heavily polluted), and the maximum elements $C_1$ and $C_8$ and the minimum element $C_6$ are obtained. There is no connection line between the adjacent elements $C_4$ and $C_7$ as well as $C_3$ and $C_1$, and it is considered that they have at least one evaluation index with opposite attributes. There are connecting lines between adjacent elements such as $C_7$ and $C_1$ as well as $C_2$ and $C_3$, indicating that the attribute values of all evaluation indexes increase synchronously. The final sample clustering results are shown in Table 2, and the attribute values of clustering evaluation indexes are shown in Table 3.

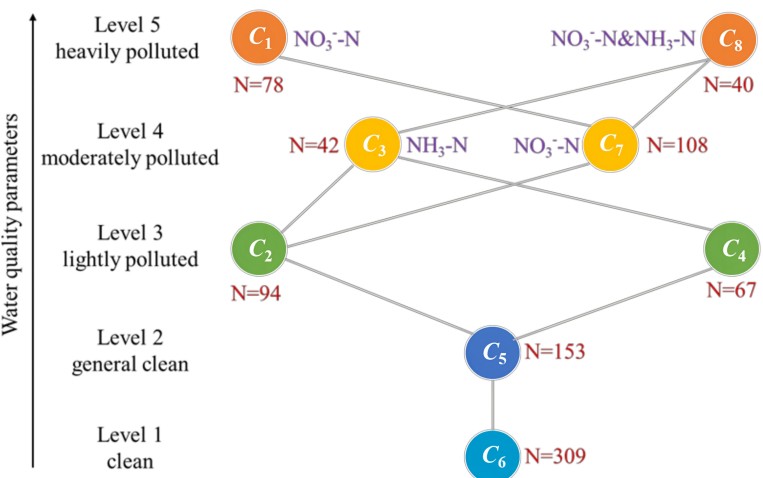

**Figure 4.** Schematic depicting the Hasse diagram where $C_{1,2,...,8}$ represents elements used for SOM clustering, N represents sample number, and the purple text represents the evaluation indices driving element pollution.

**Table 2.** Clustering results of SOM and HDT.

| Level | Element | N | Neuron Number | Corresponding Samples (Section-Month) |
|---|---|---|---|---|
| Level 1 | $C_6$ | 309 | 7,8,9,10,11,12,13,14,15,23,24,25,26,27,28,29, 30,39,40,41,42,43 | JM19-3, JM23-3, JM23-4, JM4-5, JM22-8, J14-10, J15-10, JM13-2, JM3-2, JM3-3, J2-5, JM3-5, T5-8, T5-7, T9-10,... |
| Level 2 | $C_5$ | 153 | 5,6,20,21,35,36,37,38,44,45,51,52,54,55,56, 57,58,60,67,68,69,70,71 | J3-1, T5-2, J3-3, JM18-4, JM20-4, J29-6, J30-8, J18-9,J31-11, J18-3, JM20-3, JM19-4, J3-7, J13-8, J3-12, J18-1, JM18-1,... |
| Level 3 | $C_2$ | 94 | 50,59,66,72,73,74,75,82,83,84,85,86,88,97, 98,99,112,113,114 | JM9-12, T12-12, T9-1, JM22-2, T9-2, T1-3, T1-5, JM5-6, T1-11, T9-11, JM20-1, JM19-2, JM19-12, JM5-5, J13-1,... |
| | $C_4$ | 67 | 2,3,4,18,19,33,49,65,80,95,109,110,122,124, 125,126,127,137,138,139,140,142 | J19-8, T11-12, J3-5, J18-4, J30-6, T15-6, J25-7, J4-7, J18-11, T12-9, T13-11, J5-12, JM9-2, J5-7, J6-9, T11-11, J19-1,... |
| Level 4 | $C_3$ | 42 | 1,16,17,31,32,48,63,64,79,93,94,107,108,121 | J25-12, T11-4,J24-1, J30-1, J7-7, T11-2, T4-3, J19-4, J30-2,... |
| | $C_7$ | 108 | 87,89,90,100,101,102,103,104,105,116,117, 118,129,130,131,143,144,145 | JM17-6, JM24-8, JM20-10, JM24-12, T2-2, T6-7, T8-7, T5-9, T7-10, T2-3, T2-4, J5-10, J3-10, J6-6, J6-2, J6-3, JM23-9,... |
| Level 5 | $C_1$ | 78 | 119,120,132,133,134,135,146,147,148,149,150 | T16-4, T16-5, T16-6, T16-9, T16-10, J29-1, J30-9, J7-3,... |
| | $C_8$ | 40 | 46,47,61,62,76,77,78,91,92,106 | J8-4, JM10-5, T13-4, T13-5, J25-4, JM10-2, T17-3, T17-6,... |

**Table 3.** Attribute values of evaluation indices in clustering results (unit: mg/L).

| Level | Element | TN | | $NH_3$-N | | $NO_3^-$-N | | $NO_2^-$-N | | DON | |
|---|---|---|---|---|---|---|---|---|---|---|---|
| | | ave | SD | ave | SD | ave | SD | ave | SD | ave | SD |
| Level 1 | C6 | 0.98 | 0.62 | 0.35 | 0.34 | 0.46 | 0.32 | 0.03 | 0.02 | 0.14 | 0.10 |
| Level 2 | C5 | 1.65 | 0.42 | 0.50 | 0.29 | 0.84 | 0.24 | 0.05 | 0.01 | 0.26 | 0.08 |
| Level 3 | C2 | 2.27 | 0.59 | 0.56 | 0.30 | 1.24 | 0.35 | 0.07 | 0.02 | 0.39 | 0.11 |
| | C4 | 2.87 | 0.69 | 1.45 | 0.42 | 1.04 | 0.29 | 0.06 | 0.02 | 0.33 | 0.09 |
| Level 4 | C3 | 4.32 | 1.10 | 2.34 | 0.83 | 1.44 | 0.41 | 0.08 | 0.02 | 0.45 | 0.13 |
| | C7 | 3.05 | 0.74 | 0.73 | 0.49 | 1.69 | 0.40 | 0.09 | 0.02 | 0.53 | 0.13 |
| Level 5 | C1 | 4.63 | 1.33 | 0.96 | 0.82 | 2.68 | 0.84 | 0.15 | 0.05 | 0.85 | 0.26 |
| | C8 | 6.91 | 1.63 | 3.90 | 1.45 | 2.20 | 0.56 | 0.12 | 0.03 | 0.69 | 0.18 |

The advanced relationships among nitrogen properties (mass concentration) can be analyzed by determining the relationship between different elements. Figure 4 shows the elements ($C_i$) in each level of the Hasse diagram. $C_8$ and $C_3$ represent heavy $NH_3$-N pollution, while $C_1$, $C_7$, and $C_8$ represent heavy $NO_3^-$-N pollution. The nitrogen attribute values of $C_6$ and $C_5$ were low. Nitrogen pollution gradually increased from Level 1 to Level 5; however, the nitrogen attribute values between elements did not increase with a rise in level (Table 3). Specifically, Level 1 contains $C_6$ whose nitrogen attribute values are low. Level 2 contains $C_5$, which is more nitrogenous than Level 1. Level 3 contains $C_2$ and

$C_4$, and its nitrogen properties are more profound than those at Level 2. Level 4 includes $C_3$ and $C_7$; $C_3$ shows higher TN and $NH_3$-N values, $C_7$ has higher $NO_3^-$-N values, and $C_3$ exhibits higher nitrogen attributes than those of the samples at Level 3. However, $C_7$ only increased the nitrogen attribute of $C_2$ at Level 3, which was lower than the $NH_3$-N value in $C_4$. Level 5 contains $C_1$ and $C_8$, which exhibit high nitrogen attribute values. The $NO_3^-$-N pollution of $C_1$ is dominant, while the $NH_3$-N pollution of $C_8$ is more prominent. $C_1$ only has an advanced relationship with $C_7$ at Level 4. The $NH_3$-N attribute of $C_3$ at Level 4 is higher than that of $C_1$, while the nitrogen attribute of $C_8$ is higher than all elements at Level 1 to Level 4.

## 4. Discussion

### 4.1. Comprehensive Evaluation of Nitrogen Pollution

The nitrogen pollution of rivers in Chengdu, which is mainly driven by nitrate nitrogen, has been concentrated in the middle and lower reaches. Figure 5 shows the number and proportion of samples during the high and low water periods as well as the upper, middle, and lower reaches of the hierarchical clustering results. The nitrogen pollution at the upper, middle, and lower reaches in Chengdu changed significantly compared to the variations in nitrogen pollution in the high and low water periods. Samples that were moderately and heavily polluted accounted for 30.1% of the total samples, indicating that nitrogen pollution is still prominent. With increasing pollution levels, the proportion of dry season samples increased to 57.0%, the proportion of upstream samples decreased significantly, and the proportion of downstream samples increased significantly. The upstream samples that were moderately and heavily polluted accounted for only 14.9% of the total samples, whereas the proportion of downstream samples was 85.1%. Samples subjected to $NH_3$-N pollution were dominant in the middle reaches, and there were no upstream samples. Meanwhile, samples in the dry season were more than double the samples in the wet season. For $NO_3^-$-N pollution, the proportion of downstream samples was approximately 50%, and the sample size of $C_1$ in the wet season and dry season was similar. The number of $C_7$ samples in the wet season was more than that in the dry season, while contrasting results were observed for $C_8$ because of the significant $NH_3$-N pollution. Samples subjected to low-level nitrogen pollution were mainly observed in the middle and upper reaches, and the number of samples in the wet and dry seasons was equivalent. Overall, the nitrogen attribute values of most samples were low, and the number of samples affected by $NO_3^-$-N (25.4%) was much more than that affected by $NH_3$-N (9.2%).

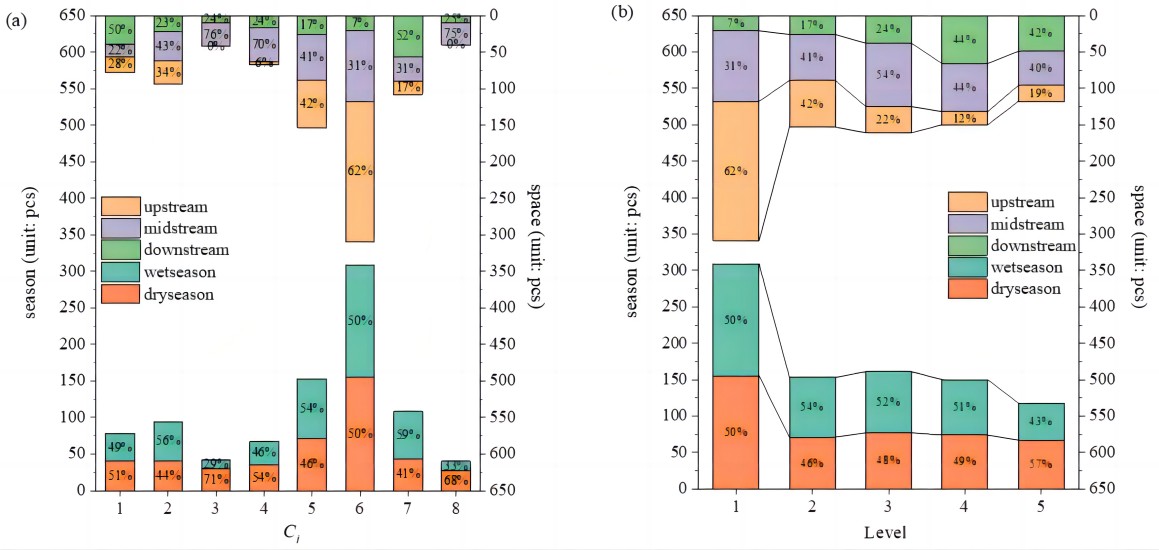

**Figure 5.** Number of samples and proportion of each cluster in rainy and dry seasons as well as upstream, midstream, and downstream.

The nitrogen pollution characteristics in the three basins tended to be slightly different. The degree of nitrogen pollution in the Tuo River Basin was greater than that in the other two basins. The proportion of clean samples was only 1.0%, and that of heavily polluted samples was 32.0%. Meanwhile, the proportion of clean samples in Jinma and Jin River Basin accounted for 51.0% and 40.3%, respectively. The proportion of samples affected by $NO_3^-$-N and $NH_3$-N was 47.4% and 14.5% in Tuo River Basin, 16.0% and 8.0% in Jinma River Basin, and 20.0% and 6.7% in Jin River Basin, respectively. The pollution range of $NH_3$-N in Jin River Basin was low, but the pollution degree was high ($3.11 \pm 1.50$ mg/L); however, all the samples were located in the middle reaches.

### 4.2. Advantages and Disadvantages of SOMs and HDT Technology

Studies have shown that the spatial and temporal distributions of various nitrogen forms in the region are complex, and the conclusions drawn by traditional single evaluation methods are often not accurate enough. Through the unorganized information provided by SOMs, numerous samples can be preliminarily clustered. Although the results provide a qualitative evaluation of water quality, a definite ranking of pollution levels cannot be obtained. Furthermore, HDT technology can elucidate ranking relationships during clustering, is not restricted by national water quality standards, and can be used to perform any standard water quality evaluation. The preprocessing of data by SOMs addresses the problem of HDT being intolerant to 'noise' to some extent. Thus, the nitrogen pollution evaluation conducted using SOMs and HDT is friendly and reliable.

Previous studies have concluded that by utilizing both SOMs and HDT, water pollution evaluation can be realized by imaging the water surface. Tsakovski et al. [12], Liu et al. [20], and Voyslavov et al. [18,19] used binomial technology to analyze the temporal and spatial characteristics of surface water pollution in Struma River, Mudan River, and Maritsa River, respectively. However, all these studies relied on local surface water standards for manual grading. In contrast, the present study employed SOMs and HDT to perform visual nitrogen pollution evaluation without utilizing any water standards. The results elucidated the spatial and temporal characteristics of nitrogen pollution in rivers, while providing another method for formulating water quality standards to better serve local water environment management.

Although the proposed method clearly exhibits advantages for evaluating surface water monitoring results, this study judges its reliability based on only the consistency of results. Since it only utilizes spatial and temporal analysis results of nitrogen forms, substantive evidence is lacking. The water quality evaluation parameters only include nitrogen-related indicators; although there is a significant correlation between these parameters, a certain deviation is also observed in the characterization characteristics. Furthermore, when using DART software for HDT analysis, it is still necessary to manually set the equivalence class samples, which is not ideal.

## 5. Conclusions

Nitrogen pollution in the rivers of Chengdu, which can be divided into five levels, is prominent and mainly driven by nitrate nitrogen. To further improve the water environment quality, controlling nitrate nitrogen pollution is key. The nitrogen pollution in the Tuo River Basin is more prominent. Meanwhile, the range of ammonia nitrogen pollution in Jin River Basin is low, but the pollution degree is high. The evaluation results obtained using SOMs and HDT are consistent with the actual situation, and thus can be used for evaluating nitrogen pollution in other rivers.

Furthermore, the evaluation of nitrogen pollution in river waters based on SOM and HDT is not restricted by water quality standards. The proposed method can be used for visual clustering and sorting, with the output results being clear and reliable. In the future, the credibility of this method can be improved and the software application development can be optimized to reduce manual operation, which will help promote its practical applicability for environmental management.

**Author Contributions:** Conceptualization, Y.D.; methodology, Y.D.; software, Y.D.; validation, Y.D., S.Y. and X.Z.; formal analysis, Y.D.; investigation, S.Y., L.O. and C.L.; resources, Y.W. and Y.D.; data curation, Y.D.; writing—original draft, Y.D.; writing—review & editing, Y.D.; visualization, Y.D.; supervision, Y.W.; project administration, Y.W.; funding acquisition, Y.D. All authors have read and agreed to the published version of the manuscript.

**Funding:** This research was jointly funded and analytically supported by the National Science Foundation of China (41473013 and 41627802) and the Water Pollution and Technology Foundation Project of the Chengdu Ecological Environment Bureau entitled 'Source Analysis of Surface Water Pollutions in Chengdu City'.

**Institutional Review Board Statement:** Not applicable.

**Informed Consent Statement:** Not applicable.

**Data Availability Statement:** The data presented in this study are available on request from the corresponding authors.

**Acknowledgments:** We are grateful to the 'Chengdu Institute of Environmental Protection' and 'Evaluation and Utilization of strategic Rare Metals and Rare Earth Resource Key Laboratory of Sichuan Province' for sampling and testing.

**Conflicts of Interest:** The authors declare no conflict of interest. The funders had no role in the design of the study; in the collection, analyses, or interpretation of data; in the writing of the manuscript, or in the decision to publish the results.

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
