# Peer review of "Evaluating Surface Water Nitrogen Pollution via Visual Clustering in Megacity Chengdu"

_water, doi:10.3390/w15112113_

Round 1

Reviewer 1 Report

1. Abstract is not informative, Add key findings of the present research 

2. Lack of literature review in introduction section 

3. Study focused nitrogen concentration in the specific study region, Add the landuse and land cover of the study region, 

4. Highlight the agriculture field in the study region, Also add the other source of nitrogen in surface water

5. Result interpretation is good, Fig.5 Is not clear. Describe it more in detail

6. Also discussion section is weak than the result section, Add the more discussion about the study results 

7. Revise the conclusion based on the correction carried out in the main text  

Minor editing of English language required

Reviewer 2 Report

I enjoyed reading the manuscript titled “Evaluating surface water nitrogen pollution via Visual clustering in megacity Chengdu". The study aims to offer chemometric expertise for comprehensively evaluating the nitrogen pollution in the river waters of Chengdu and provide a basis for local environmental legislation.

The manuscript is clear, relevant for the field. The data represent understandable documentation of the research problem. The results of the manuscript are reproducible based on the details provided in the „Chemometrics” section. The conclusions are consistent with the performed tests and the arguments presented. The construction of the manuscript is well organized.

I am convinced that the article will be interest to the scientific community and after considering the minor suggestions, I recommend publication. The paper has interesting results and will make a useful contribution to Water.

·       Line 87 - "...21 physical and chemical indicators of the samples were tested" - list what indicators, please.

·       For an easier location of the research area, please add on the Figure 1 the names of towns and rivers and all the names that are in section 2.1. Please improve the readability of the maps on the left in figure 1 (add names and scale).

·       In my opinion, the value of the work would be increased by tables containing physicochemical elements interpreted in the following. I suggest adding them as extreme value range and basic stat.

Author Response

Suggestion 1. Line 87 - "...21 physical and chemical indicators of the samples were tested" - list what indicators, please.

   Thank you for this excellent suggestion. I have changed 'The first working day of each month was used for sampling, and 21 physical and chemical indicators of the samples were tested.' to 'The first working day of each month was used for sampling, and 21 physical and chemical indicators (flow, pH, DO, temperature, EC, CODMn, BOD5, TN, NH3-N, NO3N, NO2N, DON, TP, PO43-, K+, Na+, Ca2+, Mg2+, Cl-, SO43-, CO3-) of the samples were tested.' 

Suggestion 2.  For an easier location of the research area, please add on the Figure 1 the names of towns and rivers and all the names that are in section 2.1. Please improve the readability of the maps on the left in figure 1 (add names and scale).

    I have modified the left in Figure 1 (attachment figure 1). However, when I add names of towns and rivers, I found the figure being so chaos (attachment figure 2). Thus I decided to use the figure of attachment figure 1.

Suggestion 3.  In my opinion, the value of the work would be increased by tables containing physicochemical elements interpreted in the following. I suggest adding them as extreme value range and basic stat.

    I think it's a good opinion. Because the table is wider than the layout, there is no increase in data extremes. I have tried to modify the table form, and haven't found better layout. I think the mean and standard deviation can already explain the overall situation of the data, there is no need to list all the data characteristics. And the extreme value of the data is not the focus of the article.

    Thank you very much for your affirmation of my research. Thank you very much for your comments on my paper.

Round 2

Reviewer 1 Report

Revise the conclusion section and add key findings here 

Refer: https://doi.org/10.3390/w15030601 and https://doi.org/10.1007/s12517-022-09553-x

Fine